# The Role of Venetoclax in Relapsed/Refractory Acute Myeloid Leukemia: Past, Present, and Future Directions

**DOI:** 10.3390/bioengineering10050591

**Published:** 2023-05-13

**Authors:** Matteo Piccini, Francesco Mannelli, Giacomo Coltro

**Affiliations:** 1Hematology Department, Azienda Ospedaliero-Universitaria Careggi, 50134 Florence, Italy; 2Department of Experimental and Clinical Medicine, University of Florence, 50134 Florence, Italy; 3CRIMM, Center for Research and Innovation of Myeloproliferative Neoplasms, Azienda Ospedaliero-Universitaria Careggi, 50134 Florence, Italy

**Keywords:** acute myeloid leukemia, hypomethylating agents, relapsed/refractory, venetoclax

## Abstract

Relapsed and/or refractory (R/R) acute myeloid leukemia (AML) is hallmarked by dramatic prognosis. Treatment remains challenging, with allogeneic hematopoietic stem cell transplantation (HSCT) as the only curative option. The BCL-2 inhibitor venetoclax (VEN) has proven to be a promising therapy for AML and is currently the standard of care in combination with hypomethylating agents (HMAs) for newly diagnosed AML patients ineligible for induction chemotherapy. Given its satisfactory safety profile, VEN-based combinations are increasingly being investigated as a part of the therapeutic strategy for R/R AML. The current paper aims to provide a comprehensive review of the main evidence regarding VEN in the setting of R/R AML, with a specific focus on combinational strategies, including HMAs and cytotoxic chemotherapy, as well as different clinical settings, especially in view of the crucial role of HSCT. A discussion of what is known about drug resistance mechanisms and future combinational strategies is also provided. Overall, VEN-based regimes (mainly VEN + HMA) have provided unprecedented salvage treatment opportunities in patients with R/R AML, with low extra-hematological toxicity. On the other hand, the issue of overcoming resistance is one of the most important fields to be addressed in upcoming clinical research.

## 1. Introduction

Relapsed and/or refractory (R/R) acute myeloid leukemia (AML) represents one of the most challenging scenarios in hematology, with a 5-year survival of only 10% [1]. Some parameters can influence the probability of response to salvage therapy, as well as long-term survival, restricted to the delivery of allogeneic hematopoietic stem cell transplantation (HSCT) [2]. In candidate patients, the latter is the only reliable option with curative potential.

Historically, salvage has been based on high-dose chemotherapy, aiming to obtain a response that is supposed to facilitate the efficacy of the immunologic graft-versus-leukemia effect. Although pursuing a response is a rational strategy, salvage chemotherapy is often featured by remarkable toxicity, which can impair the feasibility of HSCT in a significant fraction of patients. According to this view, the central role of HSCT even questions the opportunity of an attempt to achieve a second remission [3]. In a different clinical context, namely patients harboring *FLT3* mutations, the advantages of an effective bridging strategy with a relatively safe profile are clearly expressed by the results obtained with gilteritinib with respect to conventional chemotherapy, both in terms of non-relapse mortality and relapse prevention [4,5,6].

On the other hand, the role of disease status in the outcome of HSCT is established, and the presence of measurable residual disease (MRD) before the procedure markedly increases the probability of relapse [7].

Overall, the clinical management of R/R AML patients requires a therapeutic approach capable of providing rapid achievement of a high-quality response, together with a low burden of toxicity, as a bridge to HSCT. That clearly pertains to patients eligible for curative intent. Needless to say, the prognosis of elderly or unfit R/R patients is even more disappointing, so the approach to this patient subset is generally focused on different aims, such as disease control, transfusion dependency reduction, and improvement of quality of life.

The induction of apoptosis by BCL-2 inhibition with venetoclax (VEN) is clearly emerging among the most promising therapeutic modalities in AML [8]. Having been established as the standard of care for untreated AML in elderly and/or unfit patients [9], the low extra-hematological toxicity anticipates the suitability of VEN-based combinations as part of the therapeutic strategy in R/R AML, especially in view of the crucial role of HSCT.

This review aims to summarize the main evidence available on VEN in the context of R/R AML patients, what is known about drug resistance mechanisms, and the room for future development with a view to improving results.

## 2. From Preclinical Evidence to Early Clinical Experiences with Venetoclax in AML

The B-cell lymphoma 2 (BCL-2) family of proteins regulates mitochondrial outer-membrane permeabilization (MOMP), the crucial event in the regulation of the apoptotic process [10]. BCL-2 antagonizes the activation of apoptosis by sequestering proapoptotic family members required for MOMP. A relevant role of BCL-2 has been implicated in the survival of AML cells. This has supported the development of oral BCL-2 inhibitors in diverse clinical contexts.

Venetoclax is a small molecule that acts as a potent, selective inhibitor of BCL-2. It is a first-in-class BH3 mimetic that acts by mimicking the BH3 domain of all proapoptotic BCL-2 family proteins. The drug displaces BCL-2 from binding to a set of proteins, specifically the pore-forming proteins BAX and BAK, thereby making them capable of permeabilizing the mitochondrial membrane. The dependence of the cell on the effect of VEN relies on the amount of BCL-2 that sequesters proapoptotic proteins, a sort of functional state of BCL-2, otherwise called “primed”, the entity of which is predictive of the ability of VEN to induce MOMP [11].

In 2006, Konopleva et al. [12] provided compelling preclinical evidence that BCL-2 inhibition may yield meaningful therapeutic potential in AML. By exposing AML cell lines and primary blasts from R/R AML patients to the BCL-2/BCL-X_L_/BCL-W inhibitor ABT-737, they were able to demonstrate the disruption of BCL-2/BAX association, resulting in the freeing up of BAX and BIM, which, in turn, leads to cytochrome c release, caspase activation, and finalization of the apoptotic cascade. The antileukemic potential of ABT-737 was subsequently confirmed in murine models of AML [13].

In 2014, Pan et al. [14] reported on the efficacy of VEN (ABT-199) on multiple AML cell lines and patient-derived xenografts at nanomolar concentrations. Interestingly, the cytotoxicity of VEN was largely independent of mutational status and was maintained on chemorefractory primary AML blasts.

A phase II study assessed the clinical activity of VEN monotherapy (800 mg daily) in 32 R/R AML patients [8]. The study population consisted of heavily pretreated patients with a median age of 71 years (range, 19–84) and was significantly enriched with high-risk cytogenetic/molecular features. While safety signals were encouraging, the single-agent activity of VEN was found to be modest, with 6/32 patients meeting the criteria for complete remission (CR) or CR with incomplete count recovery (CRi). Additionally, a disease burden reduction not qualifying for CR was observed in 19% of patients. Responses were mostly short-lived, with a median progression-free survival (PFS) of 2.5 months. The study also provided preliminary insights on possible biomarkers for VEN sensitivity. Specifically, the presence of *IDH1*/*IDH2* mutations and the lack of blast dependence on BCL-X_L_ and MCL-1 were associated with response to VEN, whereas the presence of mutations in RAS pathway genes, particularly *FLT3*-ITD and *PTPN11*, correlated with treatment failure.

Venetoclax has been shown to synergize with a number of agents already in use in AML, including hypomethylating agents (HMAs) and cytotoxic chemotherapy [15,16,17,18,19], providing the rationale for a combinational strategy.

Driven by the promising efficacy data on heavily pretreated patients and the relatively favorable toxicity profile, the clinical development of VEN in AML rapidly shifted towards the frontline setting for elderly/unfit patients, where therapeutic standards [20,21,22] (HMAs and low-dose cytarabine (LDAC)) had been providing unsatisfactory outcomes in clinical practice [23,24,25,26,27]. Moreover, the possibility of increasing treatment efficacy without excessive toxicity with an orally available drug was valuable. Clinical trials with VEN in combination with HMAs or LDAC in elderly/unfit patients in the frontline setting demonstrated striking efficacy with unprecedented overall response rates (ORRs) and prolonged overall survival (OS) and exhibited an overall manageable toxicity profile [9,28,29], marking a paradigm shift in the therapeutic approach to this patient population.

## 3. Venetoclax in R/R AML

Although VEN-based regimens have been gaining popularity in the context of salvage therapy, their positioning in the setting of R/R AML is far less clear. The clinical development of VEN in this setting has suffered critically from the lack of interventional, multicenter clinical trials, not to mention the absence of randomized studies comparing VEN-based regimens to conventional strategies. Nonetheless, as the number of patients treated with VEN in the frontline setting in combination with HMAs or cytotoxic chemotherapy steadily increases, its role in the context of salvage regimens is bound to become more problematic.

At the present time, clinical evidence regarding VEN-based regimens in R/R AML is largely derived from small, single-arm studies and retrospective case series from real-world experiences involving markedly heterogeneous patient populations (Table 1). While all these experiences certainly provide valuable insights regarding the efficacy and safety of VEN-based regimens, the lack of high-quality data means that therapeutic decisions based more on clinicians’ experience than compelling evidence.

The following paragraphs reviews the use of VEN-based combinations for R/R AML patients in different clinical settings using different combinational strategies.

### 3.1. Venetoclax + HMAs or LDAC for R/R AML

Preclinical evidence has highlighted the synergistic effect of combining BCL-2 inhibitors with HMAs and LDAC [15,16,17,18]. The rationale for exploring the combination of VEN with such lower-intensity treatments in R/R AML lies in the possibility of achieving clinical efficacy with reasonable toxicity. This concept is of utmost relevance in the context of R/R disease, where patients have already been exposed to the toxicity of previous chemotherapy and might develop significant end-organ damage if exposed to further intensive treatment. In the case of younger/adult patients with R/R AML, the possibility of achieving a meaningful clinical response with limited toxicity is expected to translate into a higher rate of successful HSCT transition and, possibly, lower rates of transplant-related mortality (TRM). For elderly/unfit patients, who are not usually candidates for HSCT with curative intent, VEN in combination with lower-intensity regimens could provide an accessible platform aimed at achieving a survival benefit while preserving the quality of life. Finally, the possibility of disease control with limited toxicity could be beneficial for patients relapsing after HSCT, who are often not candidates for intensive reinduction approaches due to the persistence of transplant-related toxicities. In this setting, VEN-based lower-intensity approaches could interact positively with donor lymphocyte infusions (DLIs) or provide a bridging platform for a second HSCT.

Initial experiences with VEN + HMAs/LDAC were mainly derived from retrospective case series including heavily pretreated patients with relatively advanced median age enriched in high-risk cytogenetic and molecular features and treated outside of clinical trials (Table 1). In 2017, DiNardo et al. [30] provided a retrospective analysis of 43 patients treated off-protocol with VEN in combination with HMAs or LDAC for R/R AML (n = 39), myelodysplastic syndrome (MDS) (n = 2), or blastic plasmacytoid dendritic cell neoplasm (BPDCN) (n = 2). The median age was 68 years (range, 25–83). In the AML cohort, more than 30% of patients had received treatment for an antecedent hematological neoplasm. Most patients (84%) were treated in a second or further salvage setting, and 77% of patients had received previous HMA therapy. Almost 50% of patients had adverse cytogenetic abnormalities, and high-risk molecular features (mutations in *TP53*, *RUNX1*, and *ASXL1*) were highly represented. The overall response rate was 21% (CR + CRi, 12%; morphologic leukemia-free state (MLFS), 9%). The median OS was 3 months for the whole cohort and 4.8 months for responding patients. As previously reported [8], *IDH1/2* and *RUNX1* mutations appeared to confer a higher likelihood of response (ORR, 27% and 50%, respectively). The main adverse events (AEs) were cytopenias (mainly grade 3–4 neutropenia; 100%) and infections (mainly pneumoniae; 40%). It must be noted that 47% of patients were already receiving intravenous antimicrobials for active infections at the time of treatment initiation, not surprisingly for a heavily treatment-experienced patient cohort. These results have fostered subsequent explorations of VEN + HMAs/LDAC in R/R AML worldwide.

Aldoss et al. [31] reported on the outcomes of 33 consecutive adult patients (median age, 62 years; range, 19–81) with R/R AML treated with VEN + HMAs at City of Hope Medical Center between 2016 and 2017. Previous HMA treatment was reported in 60.6% of patients, while 39.4% had previously received HSCT. High-risk genetic features were reported in 54.5% of patients. Most patients (31/33) received VEN in combination with decitabine (DAC), 51.6% of them with a longer 10-day schedule. The median number of cycles was 2 (range, 1–10), with a relatively short median follow-up of 6.5 months. An ORR of 64% was observed (CR, 30%; CRi, 21%; MLFS, 9%). Consistent with the previous report by DiNardo et al. [30], the best response was observed after a median of two cycles. Interestingly, 53% of patients with available MRD evaluation were MRD-negative, suggesting that VEN + HMA combination may yield deep responses in a subset of patients with R/R AML. A ten-day DAC schedule was not associated with improved outcomes. While previous HSCT and HMA exposure did not negatively affect the response rate, de novo AML, the absence of high-risk cytogenetics, and the presence of *IDH1/2* mutations were associated with improved ORR. Interestingly, 67% of *TP53*-mutated and 44% of *FLT3*-mutated (TKD or ITD) patients achieved an objective response. One-year OS was 53% (73% for de novo AML patients), and median disease-free survival (DFS) was 8.9 months. Safety signals were in line with previous experiences, and most serious AEs were deemed unrelated to therapy.

These data were confirmed in a subsequent extension [32] of the original case series (n = 90; median age, 59 years; range, 18–81), with additional insights into the correlation between molecular features and response to therapy. In this larger cohort (previous HMA exposure was reported in 51% of patients), a 46% ORR was reported. Despite the limitations of the small sample size for individual genetic groups, the authors reported objective responses across all genetic subtypes. In multivariate analysis, European Leukemia Net (ELN) genetic risk was associated with differential ORR, while the presence of either *ASXL1* or *TET2* mutations was associated with better CR/CRi. In univariate analysis of mutations and genetic functional pathways, the presence of *TP53* mutations (*p* = 0.049) and alterations in chromatin-modifying genes (*p* = 0.002) adversely influenced the OS. However, in multivariate analysis, neither had an independent impact on OS.

A 10-day DAC schedule was also explored by DiNardo et al. [33] in a phase II trial conducted at the MD Anderson Cancer Center (MDACC) including 55 AML patients R/R after a median of 2 previous lines (range, 1–3); the median age was 62 (range, 43–73). A very promising ORR of 62% was reported, with 54% of MRD-negative responses. However, 18% of patients received concomitant genotype-targeted agents (FLT3 inhibitors, enasidenib, and ponatinib), which may have influenced ORR favorably.

The Mayo Clinic experience with VEN + HMAs in AML was reported by Morsia et al. [34] in 2020. This retrospective analysis included 42 patients with R/R AML (35.7% were HMA-experienced). Sixty-six percent of patients were classified into the ELN high-risk group. A 33% ORR was observed with frequent CRi (14.3%). The median number of cycles to best response was one, and the median duration of response was 2.0 months. In keeping with the City of Hope Medical Center experience [31], responses were observed across all genetic groups, but the sample size for individual genetic subsets was too small to draw conclusions. Eight patients (19%) were able to proceed to HSCT after VEN-based treatment. The median OS was 5 months (15 months for patients achieving CR/CRi; 16 months for patients proceeding to HSCT).

In 2021, our group reported its clinical experience in treating R/R AML patients with VEN combined with HMAs (azacitidine (AZA), n = 29; DAC, n = 5) or LDAC (n = 13) [35]. The median age was 56 years (range, 33–74). Notably, most patients included in our case series were primary R/R early after frontline intensive cytotoxic chemotherapy, and very few patients (n = 3) had received previous treatment with HMAs. Moreover, our case series was enriched with ELN 2017 favorable and intermediate-risk patients (70%). We observed a 55% CR + CRi rate; 50% of responses were CRi with a relatively high frequency of MRD negativity (61%). Response kinetics were in keeping with previous reports [30,31,32,33,34]. The median event-free survival (EFS) was 4.5 months, and DFS was 10.2 months, probably reflecting the relatively high percentage of CR patients proceeding to HSCT. While this study lacked a comprehensive molecular characterization of patients, traditional risk factors failed to predict CR achievement and survival outcomes. Nonetheless, the presence of *FLT3*-ITD, especially with concomitant *NPM1* mutations, was found to be associated with significantly shortened EFS and OS. A particularly dismal outcome was also noted for patients treated for post-HSCT relapse.

Owing to the lack of randomized comparisons of AZA, DAC, and LDAC as partner drugs in VEN-based regimens, the choice is often driven by clinicians’ preference and convenience, the only exception being the use of LDAC, for which available data from the frontline setting suggest [28,29] inferior activity compared to HMAs. Notably, evidence of synergism with VEN through MCL-1 downregulation is available for all three drugs [17,18,19,36].

Stahl et al. [37] reported the clinical outcomes and biological correlates of 86 R/R AML patients treated with VEN in combination with AZA (n = 35), DAC (n = 20) or LDAC (n = 27) at Memorial Sloan Kettering Cancer Center (MSKCC) between August 2016 and February 2021. The median age was 67 years (range, 29–86). Previous exposure to HMA was reported in 57% of patients, who were predictably older and predominantly allocated to VEN + LDAC. Seventeen percent of patients relapsed after HSCT. An ORR of 31% was reported for the whole cohort. Although the use of AZA was associated with significantly higher ORR and survival outcomes compared to DAC and LDAC (ORR, 49% vs. 25% vs. 15%, respectively, *p* = 0.02; median OS, 25 vs. 5.4 vs. 3.9 months, respectively, *p* = 0.003), it must be stressed that the non-randomized, non-interventional nature of the study might have introduced a fair amount of selection bias regarding the choice of partner drug. In fact, patients receiving DAC had a higher incidence of high-risk genetic features, while patients receiving LDAC were significantly older (median age, 74) and mostly R/R after frontline HMAs (88%), most likely reflecting different patient populations. Interestingly, the number of previous salvage lines (0, 1, or 2) did not seem to exert a significant detrimental effect on ORR for the whole cohort, suggesting a substantial biological difference between the antileukemic activity of VEN + HMAs/LDAC compared to conventional chemotherapy and the ability to overcome, at least in part, traditionally defined chemoresistance. However, failure of ≥3 previous lines of therapy was associated with a lower ORR (*p* = 0.04), even in the setting of VEN-based treatment. Median OS and DFS were 6.1 and 7.8 months, respectively. Fifteen patients were transitioned to HSCT after VEN-based treatment, achieving a significant OS benefit.

The effect of previous HMA exposure on the response to VEN + HMAs was assessed by Feld et al. [38] in a retrospective analysis including 44 R/R AML patients (median age, 61 years); 59.1% had previous HMA exposure. A 38.5% CR + CRi rate was reported for the whole population. Interestingly and in contrast to the MSKCC experience [37], previous HMA exposure translated to a lower ORR (exposed, 14.3%; unexposed, 66.7%).

Recently, the PETHEMA group presented the results of a multicentric, retrospective analysis of 51 R/R AML patients (median age, 68 years; range, 25–82) treated with VEN in association with AZA (n = 30), DAC (n = 15) or LDAC (n = 6) in Spain [39]. Previous HMA treatment was reported in 51% of patients, and 61% had received at least two previous lines of therapy. A rather unsatisfactory 12.4% CR + CRi rate was observed. The choice of AZA as partner drug seemed to yield a higher CR rate. The probability of achieving a CR was affected by the presence of *NPM1* mutations and mono- or biallelic *CEBPA* mutations.

In conclusion, a definitive assessment of the efficacy of VEN-based lower-intensity regimens in the context of R/R AML is extremely challenging, the main limitations being the retrospective nature of most studies, selection biases affecting the comparability of different study populations, and the frequent inclusion of patients with previous HMA exposure. Although limited clinical evidence suggests the superiority of AZA over DAC and LDAC [37,39], the lack of randomized clinical trials must be taken into consideration. The safety profile of VEN + HMAs/LDAC in R/R AML is generally regarded as manageable and overall favorable overall [28,29,30,31,32,33,34,35,37,39] compared to conventional salvage chemotherapy. The risk of tumor lysis syndrome (TLS) in AML during treatment with VEN is low, with most reported cases falling into the “laboratory TLS” definition rather than “clinical TLS” [40,41,42]. While treatment is often initiated in inpatient facilities, it has been shown that it can be safely conducted in an outpatient setting [35,42], where the availability of properly trained home care services provides invaluable support to the patients and their families [42]. In the case of R/R AML patients, hospitalization is often mandated by unresolved toxicities and profound cytopenias resulting from previous treatments.

Aside from cytopenias, an expected consequence of any form of treatment for R/R AML, infectious complications represent the main category of AE across reports. Patients with R/R AML are generally frail and carry the burden of immunosuppression resulting from previous treatments. As most of the retrospective studies examined so far included mostly heavily pretreated patients with a median age >60, it is not surprising that infectious complications were frequently reported. A definitive evaluation of the risk of infection in R/R AML is challenging, mostly due to the risk of underreporting resulting from the retrospective nature of most studies and different conduits regarding antibacterial prophylaxis, especially in the outpatient setting. In a recent update of our experience [43], only 24 febrile neutropenia episodes were observed on a total of 276 recorded VEN + HMA/LDAC cycles. We speculated that the low mucosal toxicity of VEN + HMA/LDAC might translate into a lower risk of bacterial translocation and, therefore, a relatively low incidence of febrile neutropenia events.

Invasive fungal infections (IFIs) deserve special consideration. Prolonged neutropenia in the context of induction/salvage chemotherapy has been considered an indication of mold-active antifungal prophylaxis in AML [44]. Unfortunately, azole antifungals interact with VEN via inhibition of CYP3A4 [45], increasing systemic exposure to VEN. Despite the availability of pharmacokinetic evidence and recommendations [46,47,48] for VEN dose reduction with concomitant azole administration, clinical conduits regarding antifungal prophylaxis vary greatly among different institutions. Although mainly derived from experiences with VEN + HMA as first-line treatment for AML, currently available data suggest that the rate of IFIs in patients receiving VEN + HMA is generally low (5%) [49]. However, in a report by Aldoss et al. [50], a 19% incidence of IFIs was reported among patients receiving VEN + HMA for R/R AML, with both R/R status and refractoriness to VEN + HMA independently associated with an increased risk of IFI.

### 3.2. Venetoclax + Intensive Chemotherapy for R/R AML

The demonstration of improved clinical outcomes when VEN is added to lower-intensity regimens [9,28,29] has led to the question of whether its addition to higher-intensity chemotherapy could yield higher ORRs translating into a tangible survival benefit (Table 1). Clinical trials assessing the efficacy and safety of VEN in combination with intensive chemotherapy (IC) for previously untreated AML patients have reported remarkably high CR rates [51,52,53,54]. However, combining VEN with IC comes with challenges. Specifically, the effect of VEN on the grade and duration of cytopenias appears to be particularly relevant when combined with IC, translating into a higher risk of neutropenia-related infectious complications. For example, in the MDACC trial combining VEN with FLAG-Ida [52], a protocol amendment reduced VEN exposure duration from 21 to 14 days and cytarabine dose from 2000 to 1500 mg/m^2^ following the observation of pronounced grade 3 and 4 neutropenia-related infections (including a case of typhlitis) in the original phase Ib study. Subsequent experiences with similar regimens have led to the adoption of an even shorter 7-day VEN schedule [51].

In the R/R AML cohorts included in phases Ib (n = 16; median age, 51 years; range, 20–73) and IIB (n = 23; median age, 47 years; range, 22–66) of the aforementioned MDACC trial [52], the median time to peripheral count recovery after cycle 1 was 37 days and was prolonged across all cohorts following cycle 2 despite the use of G-CSF. Extension of the cycle length over 40 days and dose reductions were eventually required for a large portion of patients proceeding after cycle 1, particularly for secondary and R/R AML patients. The CR + CRi rate for R/R patients was 67% (69% were MRD-negative), and 46% of R/R patients proceeded to consolidative HSCT with survival benefit. Febrile neutropenia and pneumonia were frequently observed (50% and 28%, respectively) and equally affected newly diagnosed and R/R patients, although bacteremia was significantly more frequent in R/R patients (46% vs. 21%, *p* = 0.04). Nevertheless, the overall safety profile in R/R AML patients was not significantly different from what is usually expected with conventional salvage regimens [55,56,57,58,59,60].

Wolach et al. [61] retrospectively analyzed the outcomes of 24 R/R AML patients (median age, 53.4 years; range, 30.1–72) treated with VEN + FLAG-Ida in Israel in a real-world setting. The median number of previous lines of therapy was one (range, 0–3), and 44% of patients had previously received HSCT. The observed CR + CRi rate was 72% (91% in the post-HSCT group); DFS and OS at 12 months were 67% and 50%, respectively. Thirty-day mortality was 12%, and 48% of patients developed bacteremia. Notably, the authors reported a relatively high (32%) incidence of IFI. Count recovery occurred at a median of 31 days (95% CI, 17.6–38.3) for platelets and 23 days (95% CI, 20–28) for neutrophils.

A retrospective single-center analysis by Shahswar et al. [62] compared the outcomes of 37 patients receiving FLA-Ida + VEN (100 mg daily for 7 days with concomitant posaconazole) with those of a cohort of 81 patients treated with FLA-Ida without VEN between 2000 and 2018. The two populations were balanced regarding median age (54 vs. 52 years, respectively), genetic features, R/R status, and previous HSCT. Patients in the FLAVIDA group had a significantly higher ORR (78% vs. 47%, *p* = 0.001), although without a significant impact on OS. Interestingly, times to count recovery did not differ significantly between the two groups.

A combination of VEN + high-dose cytarabine and mitoxantrone (HAM) was explored in the phase I/II Alliance Leukemia (SAL) Relax trial [63]. Twelve patients (median age, 57 years; range, 40–70) with relapsed AML were enrolled in the dose escalation part. The combination of VEN 400 mg daily (after a 3-day ramp-up) on days 3 to 14 + HAM was shown to be safe and effective, with 11 out of 12 patients achieving a CR/CRi.

Overall, these data provide an encouraging outlook on the efficacy of VEN in combination with IC in R/R AML. While ORRs are generally impressive, toxicity, particularly myelosuppression, is significant and requires careful consideration. Shorter VEN schedules are warranted in combination with IC, and such salvage regimens should be reserved for younger AML patients under the age of 60.

### 3.3. Venetoclax as Bridge-to-Transplant and Salvage Approach for Post-HSCT Relapse

Responses to VEN-based regimens in R/R AML are mostly short-lived in the absence of consolidative HSCT. Nonetheless, evaluating the clinical usefulness of VEN-based regimens as bridge-to-transplant platforms is problematic. First, most retrospective case series mainly include very advanced AML patients with previous exposure to multiple treatment lines, including HSCT, with extremely limited clinical options. Secondly, especially in the setting of R/R AML patients treated with VEN-based lower-intensity regimens, previous HMA exposure is a very frequent finding. Although not clearly stated by the authors, it is reasonable to assume that those were predominantly transplant-ineligible, elderly patients and/or high-cytogenetic-risk patients treated with frontline single-agent HMAs as standard clinical practice. All this considered, it is unsurprising that very few patients were reported to be able to proceed to HSCT (Table 1).

Additionally, assessing the efficacy of a VEN-based bridging strategy would require an explicit statement regarding the number of patients treated with an intention to transplant (ITT), which is very rarely found in the context of retrospective studies. In our experience [43] involving 67 R/R AML patients with a relatively low median age (58 years; range, 33–74), 39 patients (58%) were treated with VEN + HMAs/LDAC with an ITT. Ultimately, the HSCT actualization rate was 66%, and the only reason for failure to proceed with HSCT was refractory disease. This rather high rate of HSCT bridging compares favorably with other experiences, but it must be noted that our case series included mostly young/adult patients with ELN favorable/intermediate-risk R/R AML and very few post-HMA patients.

Zappasodi et al. [64] reported the outcomes of 10 R/R AML patients treated with VEN + AZA specifically as a bridge to transplant. The ORR was 60%, and all responding patients were able to transition to HSCT. Infectious complications were observed in 4/10 patients (including a case of sinonasal aspergillosis), all of which were successfully managed. In the MDACC VEN + FLAG-Ida trial [52], the transplantation rate in the R/R cohort was 46%, with a clear OS benefit for patients receiving consolidative HSCT. However, patients enrolled in the trial were predominantly young/adults and in first salvage. Although no randomized study comparing VEN-based strategies with conventional salvage chemotherapy is currently available, a propensity score-matching analysis by Maiti et al. [65] examined the outcomes of 65 patients with R/R AML treated with VEN + 10-day DAC compared to 130 IC recipients. Although this study was not powered to specifically address the question of which salvage platform might provide better HSCT bridging and the HSCT rate did not differ significantly between the two groups, overall efficacy outcomes were in favor of VEN + DAC.

Another relevant question regarding VEN-based bridging strategies is whether the addition of IC to VEN [52,63,66,67] can provide additional benefits in terms of efficacy compared to VEN + HMAs. Although reported ORRs tend to be higher for patients treated with VEN combined with IC, no randomized trials are currently available to conclusively address this question, and the patient populations involved are profoundly different in terms of demographic, clinical, and molecular features. While VEN + HMAs is an intuitively favored choice in the case of elderly R/R AML patients, it is unclear to which extent younger patients benefit from further escalation of intensity. This consideration might be particularly relevant from a safety perspective, since the use of IC is expected to be associated with a higher risk of end-organ damage, possibly jeopardizing HSCT eligibility.

The overall favorable safety profile of VEN-based regimens represents an appealing option for patients relapsing after HSCT. However, data from current scientific literature are conflicting. Several retrospective experiences with VEN-based lower-intensity regimens [35,37,43] have reported lower ORRs in the post-HSCT setting, although this observation might have been biased by the enrichment in high-risk cytogenetic/molecular features in relapsed HSCT recipient cohorts. Conversely, other experiences have reported encouraging results. Aldoss et al. [31] showed no detrimental effect of previous HSCT exposure on ORR following VEN-based salvage (previous HSCT, 46.2%; no previous HSCT, 55%; *p* = 0.73). Byrne et al. [68] reported the outcomes of 21 AML patients who relapsed after HSCT and were treated with VEN (mainly combined with HMAs). The ORR was 42.1%, and the median OS was 7.8 months, with significantly longer OS in patients achieving CR/CRi (*p* = 0.005). Infectious events were relatively frequent, as were grade 4 cytopenias. VEN dose reductions were ultimately applied to all responding patients. Similarly, Joshi et al. [69] reported a 38% ORR among 29 patients relapsing with AML following HSCT. In responders, the median OS was 403 days, and the median DFS was 259 days. The safety profile was in keeping with previous reports, and shortening of VEN exposure to 21 or 14 days was applied to mitigate the duration of cytopenias.

Zhigarev et al. investigated the immunologic landscape of T cells after exposure to VEN + HMA [70]. Treatment with HMA/VEN resulted in a greater fraction of T cells with effector memory phenotype, inhibited IFN-γ secretion by CD8+ T cells, upregulated perforin expression in NK cells, downregulated PD-1 and 2B4 expression on CD4+ T cells, and stimulated T-regulatory cell proliferation and CTLA-4 expression. Based on these findings, one could speculate that VEN might exert a beneficial effect in the context of adoptive cellular therapies for patients relapsing after HSCT. The combination of AZA with DLIs has been a commonly adopted salvage/preemptive strategy for post-HSCT relapse in AML [71], and the role of the addition of VEN in this setting was explored prospectively by Zhao et al. [72] in 26 AML patients who relapsed after HSCT. An encouraging 61.5% ORR (CR, 26.9%; PR, 34.6%) was observed. Graft-versus-host disease (GVHD) developed in six patients (23.1%), with a median time to GVHD onset of 77 days. Amit et al. [73] investigated the efficacy of DLI combined with VEN monotherapy for patients with early AML relapse after HSCT (n = 22). Treatment was generally well-tolerated, and the ORR was 50%, with a median duration of response of 135 days. The incidence of acute and chronic GVHD was 18% and 27%, respectively.

Zucenka et al. [74] retrospectively compared the outcomes of 20 patients receiving VEN in combination with LDAC and D-actinomycin (followed by VEN + DLI maintenance), with 29 patients receiving FLAG-Ida for R/R AML after HSCT. Patients receiving VEN had superior ORR (70% vs. 34%, *p* = 0.02) and OS (13.1 vs. 5.1 months, *p* = 0.032). Notably, treatment-related mortality was 0% in the VEN group and 34% in the FLAG-Ida group (*p* = 0.003), supporting the idea that VEN-based lower-intensity regimens might represent an advantageous clinical option in this patient population.

Taken together, these data support a larger-scale exploration of VEN-based salvage for post-HSCT relapse.

**Table 1 bioengineering-10-00591-t001:** Main efficacy and safety results from prospective and retrospective studies including R/R AML patients treated with venetoclax-based regimens.

Authors, Year	Study Design	Study Population	Median Age (Range)	Treatment Arms/Regimen	Efficacy Results	Relevant Safety Findings
**Venetoclax + HMAs or LDAC for R/R AML**						
DiNardo et al., 2018 [30]	Retrospective single-center	43 R/R patients with myeloid neoplasia (including 39 with AML)	68 years (25–83)	VEN 100–800 mg daily + DAC (n = 23), AZA (n = 8), LDAC (n = 8), or other (n = 4)	ORR = 21% CR = 5% CRi = 7% MLFS = 9%	Grade 3–4 neutropenia; grade 3–4 infections (pneumonia, bacteremia, cellulitis, IFI, and urinary tract infections)
Aldoss et al., 2018 [31]	Retrospective single-center	33 R/R AML patients	62 years (19–81)	VEN 400 mg daily + DAC (n = 31) or AZA (n = 2)	ORR = 64% CR = 30% CRi = 21% MLFS = 12%	Neutropenic infections (sepsis, pneumonia, colitis, and diarrhea)
Aldoss et al., 2019 [32]	Retrospective single-center	90 R/R AML patients	59 years (18–81)	VEN + DAC (n = 81) or AZA (n = 9)	ORR = 46% CR = 26% CRi = 20%	NA
DiNardo et al., 2020 [33]	Phase II single-center	168 AML patients (including 55 R/R AML)	62 years (43–73)	VEN 400 mg daily + 10-day DAC	All patients: ORR = 74% CR/CRi = 61% Median DOR = NR - R/R AML patients: ORR = 62% CR = 24% CRi = 18%, MLFS 18% Median DOR = 16.8 months	Grade 3–4 neutropenia-associated infections (6 grade 5); febrile neutropenia
Morsia et al., 2020 [34]	Retrospective single-center	42 R/R AML patients (post-HSCT excluded)	64.5 years (18–79)	VEN 100 mg daily + DAC (n = 35) or AZA (n = 8)	CR = 19% CRi = 14.3% Median OS = 5 months	Infections (85.7%), IFI (9.5%), and heart failure (19%)
Piccini et al., 2021 [35]	Retrospective single-center	47 R/R AML patients	56 years (33–74)	VEN 400 mg daily + AZA (n = 29), LDAC (n = 13), or DAC (n = 5)	CR + CRi = 55% (16% MRD-negative) Favorable outcome = Median OS = 10.7 months Bridge to HSCT = 54% (13/24)	Myelosuppression (100%), including grade 4 neutropenia (100%), grade 4 thrombocytopenia (95.7%), and grade >3 anemia (95.7%); febrile neutropenia; and infections (grade 2, n = 10; grade 3, n = 4; grade 4, n = 2)
Stahl et al., 2021 [37]	Retrospective single-center	86 R/R AML patients	67 years (29–86)	VEN 400/600 mg daily + AZA (n = 35), LDAC (n = 27), or DAC (n = 20)	ORR = 31% (49% vs. 15% vs. 25%) ^a^ CR = 14% (26% vs. 7% vs. 0%) ^a^ CRi = 10% (11% vs. 4% vs. 20%) ^a^ MLFS = 7% (11% vs. 4% vs. 5%) ^a^ Median DOR = 7.8 months Median OS = 6.1 months	NA
Feld et al., 2021 [38]	Retrospective single-center	44 R/R AML/MDS (39 AML, 5 MDS)	61.5 years	VEN 400 mg daily + AZA or DAC	ORR = 38.5% CR = 12.8% CRi = 25.6% Median DOR = 6.5 months Median OS = 8.1 months. Bridge to HSCT = 20.5% (8/39)	Grade 3 infections (59.1%), neutropenic fever (46.5%), and persistent neutropenia (71.8%)
Labrador et al., 2022 [39]	Retrospective multicenter	51 R/R AML patients	68 years (25–82)	VEN 400/600 mg daily + AZA (n = 30), DAC (n = 15), or LDAC (n = 6)	ORR = 22.9% CR = 10.4% CRi = 2% Median OS = 104 days Bridge to HSCT = 20.5% (8/39)	Neutropenic fever (53%) and bleeding (10%)
**Venetoclax + intensive chemotherapy for R/R AML**						
DiNardo et al., 2021 [52]	Phase Ib/II single-center	68 AML patients (including 39 R/R AML)	46 (20–73)	VEN 200/400 mg daily + FLAG-Ida	All patients: ORR = 82% CR = 53% CRh = 15% CRi = 7% MLFS = 4% Median DOR = NR 12-month OS = 70% - R/R AML patients: ORR = 72% CR = 44% CRh = 13% CRi = 14% MLFS = 2% Median DOR = 6-NR 12-month OS = 38–68%	Grade 3–4 AEs occurring in ≥10% of patients: febrile neutropenia (50%), bacteremia (35%), pneumonia (28%), and sepsis (12%)
Wolach et al., 2022 [61]	Retrospective multicenter	25 AML patients (including 24 R/R AML)	53.4 years (30.1–72)	VEN 400 mg daily + FLAG-Ida	ORR = 76% CR = 40% CRi = 32% MLFS = 4% 12-month OS = 50% Bridge to HSCT = 40% (10/25)	Blood stream infections (48%) and IFI (32%)
Shahswar et al., 2022 [62]	Retrospective single-center	37 R/R AML patients	54 years	FLAVIDA (VEN 100 mg daily + FLA-Ida)	ORR = 78% CR = 54% CRi = 5% Median OS = 12 months Bridge to HSCT or DLIs = 81%	Bacteremia (27%), sepsis (11%), and fungal pneumonia (11%)
Röllig et al., 2022 [63]	Phase I/II multicenter	12 R/R AML patients	56 years (40–70)	VEN 400 mg daily + HAM	CR + CRi = 92% Bridge to HSCT = 45% (5/11)	57 grade 3 AEs (37% of infectious origin)
**Venetoclax as bridge-to-transplant and salvage approach for post-HSCT relapse**						
Zappasodi et al., 2021 [64]	Retrospective single-center	10 R/R AML patients (2/10 with prior HSCT)	53 years (23–67)	VEN 400 mg daily + AZA	ORR = 60% CR = 40% CRi = 10% MLFS = 10% Median OS = 8.9 months Bridge to HSCT = 70% (6 responders, 1 non-responder)	Myelosuppression, including prolonged grade 3–4 neutropenia (100%); bacterial infections (30%); and IFI (10%)
Shahswar et al., 2020 [66]	Retrospective single-center	13 R/R AML patients (6/13 with prior HSCT)	49 years (18–62)	FLAVIDA (VEN 100 mg daily + FLA-Ida)	ORR = 69% CR = 54% Cri = 15% Median OS = NR 6-month OS = 76% Bridge to HSCT = 69% (9/13) Post-salvage DLIs = 15%	Grade 3–4 neutropenic fever (77%); Gram-negative bacteremia (23%); and grade 3–4 neutropenia, anemia, and thrombocytopenia (100%)
Abaza et al., 2023 [67]	Retrospective single-center	17 AML patients (including 7 R/R AML)	48 years (21–68)	VEN 400 mg daily + FLAG-Ida	R/R AML patients: ORR = 100% CR = 57% CRi= 14% MLFS = 14% Median OS = 6.2 months Bridge to HSCT = 57% (4/7)	NA
Byrne et al., 2020 [68]	Retrospective single-center	21 patients relapsing with AML post HSCT (primary diagnosis: AML, n = 16; MDS, n = 3; CMML, n = 1; PMF, n = 1)	64.5 years (34.5–73.7)	VEN 400–600 + AZA (n = 12), LDAC (n = 5), or DAC (n = 4)	ORR = 47% CR = 29.4% CRi = 17.6% Median OS = 7.8 months Median OS = NR for responders	All-grade infectious events (61.9%), bacterial pneumonia (33%), suspected fungal pneumonia (19%), and oral infection (9.5%)
Joshi et al., 2021 [69]	Retrospective single-center	29 AML patients in relapse post HSCT	58 years (20–72)	VEN + DAC (n = 18), AZA (n = 8), LDAC (n = 1), or other (n = 2)	ORR = 38% CR + CRi = 27.5% Median DOR = 7 months Median OS = 79 days (403 and 55 in responders and non-responders, respectively)	Grade 3–4 neutropenia (69%), grade 3–4 thrombocytopenia (65.5%), infections (55%), bacteremia (34.4%), neutropenic fever (17.2%), and fungal infection (3.4%)
Zhao et al., 2022 [72]	Clinical trial single-center	26 AML patients in relapse post HSCT	35.2 years	VEN 400 mg daily + AZA, followed by DLIs	ORR = 61.5% CRi = 26.9% PR = 34.6% Median OS = 284.5 days	Grade 3–4 neutropenia, anemia and thrombocytopenia (100%), neutropenic fever (100%), nausea and vomiting (42.3%), hyperbilirubinemia (15.4%), elevated liver enzymes (11.5%), and all-grade GVHD (23.1%)
Amit et al., 2022 [73]	Retrospective multicenter	22 AML patients in relapse post HSCT	65 years (43–75)	VEN 400 mg daily monotherapy (n = 8) or + AZA (n = 5), sorafenib (n = 5), gilteritinib (n = 3), HiDAC (n = 2), or LDAC (n = 1), followed by DLIs	ORR = 50% CR = 18% CRi = 5% MLFS = 9% Median DOR = 135 days Median OS = 6.1 months	Grade 3–4 neutropenia (73%), grade 3–4 anemia (55%), grade 3–4 thrombocytopenia (64%), grade 3–4 infection (14%), diarrhea (32%), grade 3–4 acute GVHD (5%), chronic GVHD (27%), and severe chronic GVHD (5%)
Zucenka et al., 2021 [74]	Retrospective single-center	20 AML patients in relapse post HSCT	59 years (20–71)	VEN 600 mg daily + LDAC and D-actinomycin, followed by VEN + DLIs maintenance	ORR = 75% CR = 50% CRi = 0% MLFS = 5% Median OS = 13.1 months	Febrile neutropenia (75%), bacteremia (40%), pneumonia (30%), septic shock (5%), mucositis (20%), enteritis (30%), grade 3–4 acute GVHD (10%), and TLS (5%)

Notes: ^a^ The reported percentages refer to VEN + AZA, VEN + LDAC, and VEN + DAC combinations. Abbreviations: AEs, adverse events; CMML, chronic myelomonocytic leukemia; CR, complete response; CRh, CR with partial hematologic recovery; CRi, CR with incomplete blood count recovery; DLI: donor lymphocyte infusion; DOR, duration of response; FLA-IDA, fludarabine + cytarabine + idarubicin; FLAG-Ida, fludarabine + cytarabine + G-CSF + idarubicin; GVHD, graft-versus-host disease; HAM, high-dose cytarabine + mitoxantrone; HiDAC, high-dose cytarabine; HMA, hypomethylating agent; HSCT, hematopoietic stem cell transplantation; IC, intensive chemotherapy; IFI, invasive fungal infection; LDAC, low-dose cytarabine; MDS, myelodysplastic syndrome; MLFS, morphological leukemia-free state; MRD, minimal residual disease; NA, not available; NR, not reached; ORR, overall response rate; OS, overall survival; PR, partial response; R/R AML, relapsed/refractory acute myeloid leukemia; TLS, tumor lysis syndrome; VEN, venetoclax.

### 3.4. The Role of Gene Mutations in the Prediction of Sensitivity to Venetoclax-Based Regimens

As VEN is becoming increasingly available for prescription across different countries worldwide and the therapeutic offering is broadening both in previously untreated and R/R AML with the introduction of novel molecularly targeted agents, the identification of reliable predictive biomarkers associated with sensitivity is key to rationally driving the allocation of patients to VEN-based therapies and avoiding pointless (and potentially harmful) therapeutic attempts.

Molecular lesions affecting recurrently mutated genes in AML have been reported to modulate VEN sensitivity. However, the role of single-gene lesions is often modulated by concomitant additional mutations and may vary in a context-dependent manner. Unfortunately, with very few exceptions, specific comutational patterns tend to be detected in a relatively low number of patients in most VEN clinical trials and retrospective studies in AML, which are generally not powered to investigate the predictive value of mutations/comutational patterns. Consequently, data regarding the role of single mutations are often inconclusive or even conflicting, especially in the R/R setting. Although responses to VEN-based therapies have been observed across all genetic subgroups in most studies [9,29,30,32,34], a restricted number of mutations has been consistently shown to exert a particularly significant favorable impact.

*NPM1* mutations occur in 30% of AML patients [75] and are considered predictive of improved response to induction chemotherapy [75]. Mutations in *NPM1* have been shown to correlate with higher ORR and survival outcomes in the context of VEN-based therapies in different settings. In an early phase I trial assessing the safety and efficacy of VEN + HMAs, the CR + CRi rate in 23 previously untreated *NPM1*-mutated patients was 91.5% [29]. The reported CR + CRi rate for 27 *NPM1*-mutated patients enrolled in the VIALE-A trial was 66.7% [9]. Similarly, promising efficacy data in this molecular setting emerged from the use of VEN + LDAC, with a reported CR + CRi rate of 78% [28]. Very encouraging CR + CRi rates were reported in clinical trials combining VEN with FLAG-Ida [53] (100%) or the “2 + 5” regimen (80%) [54]. Data from real-world experiences convey a similar picture [76].

Although to a lesser extent compared to the frontline setting, the presence of *NPM1* mutations has been associated with higher ORR and improved survival outcomes in R/R AML patients [35,37,43]. Recently, we reported ongoing disease-free survival exceeding 40 months in elderly patients with *NPM1*-mutated AML in first relapse receiving VEN + AZA/LDAC without consolidative HSCT [35,43]. The promising efficacy of VEN in *NPM1*-mutated AML has led to questions as to whether this approach could be used in MRD-positive patients to eradicate disease persistence and provide an advantageous platform for HSCT bridging. Tiong et al. [77] reported the outcomes of 12 patients with *NPM1*-mutant AML with MRD persistence/relapse treated with VEN + LDAC. All five patients with molecular persistence achieved complete MRD clearance and durable OS without subsequent HSCT. Additionally, six out of seven patients with molecular relapse achieved MRD eradication, and most of them transitioned to HSCT. A clinical trial conducted by the Gruppo Italiano Malattie Ematologiche dell’Adulto (GIMEMA) is currently investigating the use of VEN + AZA specifically as bridge to transplant in *NPM1*-mutated AML with MRD persistence or molecular disease recurrence (NCT04867928).

Mutations in genes encoding the isocitrate dehydrogenase isoforms 1 and 2 (*IDH1*, *IDH2*) are found in approximately 15% of AML patients [78] and have been shown to confer exquisite susceptibility to BCL-2 inhibition via cytochrome C oxidase inhibition mediated by the oncometabolite 2-hydroxyglutarate, ultimately resulting in the lowering of the mitochondrial threshold for the triggering of apoptosis [79]. The favorable impact of *IDH1/2* mutations on response to VEN, even as a single agent [8], has been reported in several clinical trials and real-world experiences both in the frontline [9,28,29,76] and R/R settings [30,35,37,43]. Overall, *IDH1/2* mutations are associated with higher ORR and longer DFS and OS, especially in previously untreated patients. Interestingly, in R/R AML, some authors have reported less of an impact for *IDH1/2* mutations, with no significant differences in outcomes compared to unmutated patients. [32,34,39] As the use of IDH1 and IDH2 inhibitors available for clinical use in R/R AML is associated with similar ORRs [80,81,82,83], the current positioning of VEN-based regimens in this population warrants further investigations.

Data regarding the beneficial impact of other mutations are difficult to interpret due to the low number of patients harboring the same genetic lesions in most study populations. Consequently, it is often not possible to establish whether specific mutations have an actual impact on clinical outcomes per se or rather in a comutational, context-dependent manner. For example, mutations affecting the epigenetic modifier genes (i.e., *DNMT3A* and *TET2*) [32,34] and the spliceosome machinery (i.e., *SRSF2*) [84] were shown to correlate with higher ORR. However, such lesions are often found in comutational contexts including mutations in genes with a known favorable impact on VEN sensitivity, such as *NPM1*, *IDH1*, and *IDH2* [84,85]. Moreover, some mutations (i.e., *TET2*) may be associated with improved susceptibility to HMA rather than VEN [86].

Overall, it is quite evident that mutations alone are not sufficiently accurate in predicting VEN sensitivity or resistance. Newer predictive tools not based on traditional cytogenetic/molecular biomarkers, such as BH3-profiling [8], might provide clinicians with useful information that might eventually drive therapeutic decisions if prospectively validated in a clinical setting.

## 4. Mechanisms of Resistance to VEN and Specific Approaches to Overcome Them

### 4.1. Mechanisms of Venetoclax Resistance

AML patients who relapse after VEN-based treatment are characterized by extremely poor outcomes [87]. VEN resistance can be mediated by heterogeneous mechanisms at the individual cell level, patterns that have not yet been fully elucidated. This scenario is even more complicated if considered in a polyclonal context, with subclones emerging under the selective pressure exerted by treatment [87]. Resistance to VEN can also be primary, i.e., intrinsically due to a scarce addiction of the leukemic cell to BCL-2 as an antiapoptotic machinery.

Overall, leukemic cells can evade the induction of apoptosis by VEN-based combinations mainly through two resistance mechanisms: (i) impairment of VEN binding and (ii) activation of alternative antiapoptosis pathways [88].

Among the first group, recurrent mutations in the *BCL2* gene decrease the affinity of BCL2 protein for VEN. On the other hand, mutations in the *BAX* gene can alter the capability of the relative proteins to localize on the mitochondrial membrane and, thus, to complete the pore-forming process [89].

On the other hand, several pathogenic alterations can bypass the inhibition of BCL-2, mainly converging on the upregulation of MCL-1 as the key mechanism of resistance to VEN [90]. On this basis, future development in the field includes pursuit of the combined use of VEN (+/− HMA) with other drugs, synergizing through inhibition of MCL-1, either selectively or indirectly, as a strategy to overcome VEN resistance.

### 4.2. The Role of Alternative BCL-2 Family Proteins

The binding to BIM and BAX by other BCL-2 family proteins is one of the mechanisms of resistance to VEN, as it shifts the balance toward MCL-1, BCL-xL, and BCL2-A1 while reducing the relevance of BCL-2 in preventing the activation of MOMP [18,91].

The centrality of MCL-1 overexpression is stated by several pieces of evidence showing the preclinical synergistic activity of VEN with some agents that could indirectly inhibit MCL-1 [92]. As an example of direct MCL-1 inhibition, the molecule S64315 is being evaluated in the early phase of a clinical trial in combination with VEN (NCT03672695). The downregulation of MCL-1 can also result from different agents sharing this effect as a final output.

Among the most promising approaches from this perspective, idasanutlin, an MDM2 inhibitor, promotes MCL-1 degradation by activating p53 [93]. The combination of VEN (600 mg daily) and idasanutlin achieved a promising ORR of 34.3% [94]. Gastrointestinal events were the most frequent AEs presenting with diarrhea, nausea, and vomiting. Cobimetinib, a mitogen-activated protein kinase (MEK) inhibitor, downregulates MCL-1 through suppression of the MAPK signaling pathway; its combination with VEN yielded an ORR of 18% in the R/R AML setting [95].

Dinaciclib and alvocidib, cyclin-dependent kinase 9 (CDK9) inhibitors, induce transcriptional downregulation of MCL-1. Clinical trials of these drugs in combination with VEN are ongoing (NCT03484520 and NCT03441555).

### 4.3. The Role of Alternative Downstream Pathways

Constitutive activation of intracellular signaling pathways is a frequent event in AML, causing wide abnormalities in the transcriptional program and, ultimately, consequences in leukemia cell proliferation and survival, affecting preferential dependence on specific antiapoptotic mechanisms [85].

***FLT3* mutations.** The presence of *FLT3* mutations is known to correlate with VEN resistance, with frequent primary refractory disease or rapid acquisition of adaptive resistance leading to transient responses [96]. Activation of pathways downstream of mutated FLT3 tyrosine kinase involves PI3K-Akt, RAS-MAPK, and STAT5 and determines overexpression or stabilization of BCL-xL and MCL-1 [97,98], both of which are responsible for reduced dependence on BCL-2. Preclinical data indicate a strong synergism between VEN and FLT3 inhibitors, collectively demonstrating potent inhibition of the aforementioned pathways and, ultimately, the downmodulation of MCL-1, thereby establishing a robust rationale for their combined use [91,99].

In a phase Ib multicenter, open-label study (NCT03625505) enrolling 61 patients with R/R AML harboring *FLT3* mutations (ITD and/or TKD), the combination of VEN and gilteritinib yielded a high rate of response (CR + CRi, 75%). Considering the clinical context, the responses were profound, with 60% of *FLT3*-mutant clearance below 10^−2^ upon molecular assessment [100]. Hematological toxicity was consistent with the known profile of VEN-based combinations. The enrolled patients were severely pretreated, having received at least two previous lines of therapy in 80% of cases.

Other combinations including FLT3 inhibitors were explored in the R/R setting, and promising results were preliminarily achieved with “triplet” therapy incorporating quizartinib with the VEN + HMA backbone (DAC). A composite response rate of 69% was observed, with a median OS of 7.1 months [101].

**Other signaling-activating mutations.** Similarly to *FLT3*, mutations in other genes encoding for tyrosine kinases or key components of signaling pathways (i.e., *RAS*, *PTPN11*, *CBL*, *KIT*, *JAK2*, etc.) are postulated to confer resistance to BCL-2 inhibition, mainly by upregulating MCL-1 and/or BCL-X_L_ [102,103]. Therefore, higher efficacy can be hypothesized in association with the combination of VEN with MCL-1 inhibitors [103].

### 4.4. The Role of TP53 Mutations

Tumor protein p53 is a transcription factor that mediates the change in the cell’s transcriptional program after the occurrence of numerous cellular stress-inducing events [104]. p53-targeted genes include several BCL-2 family members (i.e., *BAK*, *BAX*, *PUMA*, and *NOXA*), which implicate marked disruption of apoptosis regulation in *TP53*-mutated cells, the basis for the well-known chemoresistance in this AML subset [105].

*TP53*-mutated AML patients have an inferior response to induction/salvage chemotherapy, a very high relapse risk even after HSCT, and a dismal overall prognosis [106,107,108]. The benefit of adding VEN to backbone regimens in this molecularly defined subset has produced conflicting results. In the VIALE-A trial, CR + CRi incidence for *TP53*-mutated patients was 55.3% in the VEN + AZA arm and 0% in the AZA + placebo arm [9]. In the MDACC experience with VEN in combination with 10-day DAC, 35/118 patients carried a *TP53* mutation (single mutation, n = 8; multiple mutations, n = 15; mutation plus deletion, n = 12). In this patient subpopulation, CR + CRi rates were inferior compared to *TP53* wild-type patients (57% vs. 77%, *p* = 0.029), and median OS was significantly shorter (5.2 vs. 19.4 months, *p* < 0.0001) [109]. These results are comparable with historical data obtained with 10-day DAC alone [110]. Aldoss et al. [111] reported a 52% CR + CRi rate in *TP53*-mutated patients (n = 31) treated with VEN + HMAs (mainly DAC). Response rates were higher in patients treated in the frontline setting (67%), although the 38% CR + CRi rate observed in R/R AML patients is in keeping with other reports in similar settings. In the Mayo Clinic retrospective series [34], *TP53* mutations did not correlate with an inferior response rate in R/R AML patients, but the low patient number for different molecular groups in this study warrants caution. In conclusion, while ORRs with VEN-based therapies for *TP53*-mutated patients may be defined as clinically meaningful, there is a lack of compelling evidence to infer that the use of VEN in this patient population, especially in the R/R setting, is able to overcome the detrimental effect of *TP53* mutations.

Eprenetapopt (APR-246) is a novel, first-in-class small molecule that stabilizes mutant p53, thereby restoring a functional conformation of the protein [112]. The efficacy of eprenetapopt administered in combination with AZA has been tested in myeloid neoplasms bearing *TP53* mutations [113]. An increased efficacy is expected to be associated with the triple combination of eprenetapopt, VEN, and AZA, as evaluated so far in a phase I clinical trial (NCT04214860).

CD47 blockade represents an alternative therapeutic approach promising a step forward in the treatment of *TP53*-mutated AML. Magrolimab is a monoclonal antibody targeting CD47, a cell surface protein regulating prophagocytic signals in tumor cells. CD47 blockade decreases the antiphagocytic “don’t-eat-me” signal on leukemia cells and induces macrophage-mediated phagocytosis [114]. Magrolimab has shown promising response rates in combination with AZA in AML [115]. Synergism between CD47 blockade and BCL-2 targeting has been demonstrated in chronic lymphocytic leukemia [116], and it appears to be rational in AML as well. Clinical trials on the combination of VEN and magrolimab, or SIRPαFc fusion protein (TTI-622), are ongoing.

### 4.5. Other Drug Combinations

**IDH1/2 inhibitors.** Despite responding to VEN-HMA combination therapy, previous experiences have demonstrated the presence of a detectable *IDH2* mutation in a relatively relevant fraction of patients [87]. Furthermore, the occurrence of a new *IDH2* mutation has been observed in cases of relapse [37]. Therefore, the addition of an IDH inhibitor along with the VEN-HMA backbone may reasonably enhance the depth of response and prolong survival.

The triplet combination of VEN, AZA, and ivosidenib (an IDH1 inhibitor) is being tested in a phase I/II trial for *IDH1*-mutated AML patients, either naive or R/R (NCT03471260). Likewise, a combination of VEN and enasidenib (an IDH2 inhibitor) is being assessed in *IDH2*-mutated cases (NCT04092179).

**Menin inhibitors.** While *NPM1* mutations are exquisitely sensitive to VEN-induced apoptosis, concurrent *FLT3*-ITD significantly impairs the rate and durability of response [87]. Beyond the FLT3-inhibition discussed above, *NPM1*-mutated AML can be effectively addressed from another perspective with menin as the central node.

Menin is a nuclear protein that interacts with chromatin regulators and transcription factors and acts as a global regulator of transcription [117]. *NPM1*-mutated AML is dependent on binding of menin to wild-type KMT2A, making menin an actionable target in this specific disease subset [118].

Menin inhibitors have been shown to enhance the antileukemic efficacy of VEN in *NPM1*-mutant AML models [119]. A combination of the menin inhibitor DS-1594 with AZA and VEN will be evaluated in *NPM1*-mutated R/R AML in a phase I/II clinical trial (NCT04752163).

### 4.6. Conclusions and Future Directions

Venetoclax-based combinations have radically changed the therapeutic landscape of AML. In the R/R setting, which is generally characterized by dramatic prognosis, VEN has provided unprecedented salvage treatment opportunities with low extra-hematological toxicity, a crucial feature in the perspective of allogeneic HSCT for candidate patients. On the other hand, the incorporation of VEN into AML treatment has introduced a new challenging scenario represented by patients who are primary refractory or relapsing after VEN-based therapy. As VEN and HMA are likely to form the core of therapeutic management of AML in the near future, the issue of overcoming resistance is one of the most important problems to be addressed in upcoming clinical trials.

Despite the substantial amount of evidence with respect to the use of VEN in R/R AML, the current research is still subject to some limitations. These include the characteristics of the current study, as it is mostly based on retrospective, uncontrolled analysis of small populations with heterogeneous compositions, a lack of long-term data, and limited comparison to other treatments. Overall, while the results of VEN-based combinations for R/R AML are promising, further research is needed to fully understand the potential limitations and long-term outcomes of this approach.

## Data Availability

Not applicable.

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
