# Peer review of "The Role of Venetoclax in Relapsed/Refractory Acute Myeloid Leukemia: Past, Present, and Future Directions"

_bioengineering, 2023, doi:10.3390/bioengineering10050591_

Round 1

Reviewer 1 Report

Very well written and comprehensive review, but the use of the English language is poor and that makes it hard to read. In the abstract I would correct the sentence "The BCL-2 inhibitor venetoclax (VEN) has proven to be a promising therapy for AM, and is currently the standard of care for newly diagnosed AML patients ineligible for induction chemotherapy" by adding "in combination with HMA" 

Author Response

We welcome the suggestions of the reviewer. We have modified the abstract as suggested by adding “in combination with HMAs” in the third sentence of the Abstract (page 1). Moreover, the manuscript underwent extensive English revisions. We hope that our work will be appreciated.

Reviewer 2 Report

The author reviewed the current clinical research of venetoclax-based regimens in R/R AML and made a discussion on resistance mechanisms clearly, and also looked forward to future therapy strategies with venetoclax.

There are several small spelling mistakes, please correct them. For example, "AM" in line 15, "Cri" in line 217 and so on.

Author Response

We thank the reviewer for favorably reviewing our manuscript and pointing to the mistakes. Accordingly, we have revised the whole manuscript to remove or correct other spelling mistakes.

Reviewer 3 Report

The review "The role of venetoclax in relapsed/refractory acute myeloid leu-  kemia: past, present and future directions" summarizes very well the role of VEN in AML. The authors have cited the important papers in the field and have covered important aspects in this review.

Author Response

We thank the reviewer for favorably reviewing our manuscript.

Reviewer 4 Report

Matteo Piccini and collagues provide a comprehensive and very insightful review. This is easy to go through it, despite the huge amount of data. It provides a good sense of the present state-of-the-art related to the promises of Venetoclax to be key in acute myeloyd leukemia treatment, but also sheds light on the potentiel resistance issues.

Although the whole manuscript is globally fine, the conclusion may be modified, to be a little bit more conservative since data from several non- randomized and/or non-double blinded trials are reported here.

Author Response

We thank the reviewer for this excellent point. Accordingly, we have modified the “Conclusions and future directions” section to highlight the important limitations of current evidence from scientific literature, mainly due to the retrospective, non-controlled nature of most studies (page 14).